# Memory Function, Neurological, and Immunological Biomarkers in Allergic Asthmatic Mice Intratracheally Exposed to Bisphenol A

**DOI:** 10.3390/ijerph16193770

**Published:** 2019-10-08

**Authors:** Tin-Tin Win-Shwe, Rie Yanagisawa, Eiko Koike, Hirohisa Takano

**Affiliations:** 1Center for Health and Environmental Risk Research, National Institute for Environmental Studies, 16-2 Onogawa, Tsukuba, Ibaraki 305-8506, Japan; yanagisawa.rie@nies.go.jp (R.Y.); ekoike@nies.go.jp (E.K.); 2Environmental Health Sciences, Graduate School of Global Environmental Studies, Kyoto University, Kyoto 615-8540, Japan; htakano@health.env.kyoto-u.ac.jp

**Keywords:** bisphenol A, intratracheal instillation, memory function, neuroimmune biomarkers, allergic asthmatic model, mice

## Abstract

Bisphenol A (BPA) is a major constituent of plastic products, including epoxy resin containers, mobile phones, dental sealants, as well as electronic and medical equipment. BPA is recognized as an endocrine system-disrupting chemical which has toxic effects on the brain and reproductive system. However, little is known about the effects of co-exposure of BPA with allergens on the memory function and neurological as well as immunological biomarker levels. In this study, we examined the effects of intratracheal instillation of BPA on the memory function and neuroimmune biomarker levels using a mouse model of allergic asthma. Male C3H/HeJ Jcl mice were given three doses of BPA (0.0625 pmol, 1.25 pmol, and 25 pmol BPA/animal) intratracheally once a week, and ovalbumin (OVA) intratracheally every other week from 5 to 11 weeks old. At 11 weeks of age, a novel object recognition test was conducted after the final administration of OVA, and the hippocampi and hypothalami of the animals were collected after 24 h. The expression levels of the memory function-related genes N-methyl-D-aspartate (NMDA) receptor subunits, inflammatory cytokines, microglia markers, estrogen receptor-alpha, and oxytocin receptor were examined by real-time RT-PCR (real-time reverse transcription polymerase chain reaction) and immunohistochemical methods. Impairment of the novel object recognition ability was observed in the high-dose BPA-exposed mice with allergic asthma. In addition, the allergic asthmatic mice also showed downregulation of neurological biomarkers, such as NMDA receptor subunit NR2B in the hippocampus but no significant effect on immunological biomarkers in the hypothalamus. These findings suggest that exposure to high-dose BPA triggered impairment of memory function in the allergic asthmatic mice. This is the first study to show that, in the presence of allergens, exposure to high-dose BPA may affect memory by modulating the memory function-related genes in the hippocampus.

## 1. Introduction

Bisphenol A (BPA) is a major constituent of plastic products including epoxy resin containers, mobile phones, dental sealants, as well as medical and electronic equipment. The routes of entry of BPA to human body are through ingestion, inhalation and transdermal. BPA is one of endocrine disrupting chemicals and it can have an effect on brain and reproductive system. BPA has estrogenic and anti-androgenic effects on physiological and behavioral functions [1,2]. BPA may exacerbate allergic diseases such as allergic dermatitis, rhinitis, and allergic asthma [3]. Perinatal exposure to BPA not only enhanced lung inflammation in the presence of allergen [4] but also induced aggressive and anxiety behaviors in adult rats [5]; it also altered brain sexual differentiation of various brain regions in rodents [6,7].

According to epidemiological studies, exposure to endocrine-disrupting chemicals can enhance underlying asthma or allergic diseases by modulating the immune responses [8,9,10]. In a previous study, we demonstrated that exposure to toluene, a volatile organic compound, aggravated airway inflammatory responses in a mouse model of allergy by modulating the number of inflammatory cells and enhancing the plasma levels of nerve growth factor (NGF) [11]. Furthermore, long-term exposure to toluene was shown to increase the mRNA expression levels of NGF and tropomyosin receptor kinase A (TrkA) in the lungs of ovalbumin (OVA)-immunized mice [12]. Recent studies have indicated that early-life exposure to environmental chemicals may lead to lifestyle-related diseases such as obesity and diabetes mellitus and mental illnesses such as schizophrenia [13,14] in later life. The increased risk of mental illnesses, such as autism, autism spectrum disorders, schizophrenia, Parkinson’s disease, and Alzheimer’s disease, observed in offspring exposed to chemicals during early life may be due to immune system disturbances or neuroinflammation in the brain [15,16].

The prevalence of allergic diseases such as asthma, allergic rhinitis, and atopic dermatitis has increased sharply in many countries. Asthma is a chronic airway disease and is known to share common immune dysfunctions with mental disorders such as bipolar disease [17,18]. A recent study indicated a key role of Th2-mediated inflammation in the association between asthma and bipolar disease in integrin β4-KO mice [19]. Recently, we also showed that low-dose BPA exposure can aggravate allergic airway inflammation via induction of immune dysfunction and enhancement of Th2 responses [20]. It was also shown in an adult female rat model that acute, low-dose BPA exposure can interfere with estradiol-dependent consolidation of memory and alter the dendritic spine density [21].

Asthma not only affects the airways but also affects the brain. Anxiety and depression are more prevalent in asthmatic patients in comparison with healthy individuals [22,23,24]. Patients with mild asthma also show mild cognitive deficit [25], and brain MRI abnormalities have been observed in patients with mild to moderate asthma [26]. A recent study demonstrated a marked reduction of the hippocampal volume in asthmatic subjects suggesting the existence of an association between asthma and brain regional structure that triggers cognitive deficit [27]. Acute hypoxia, which can occur in asthma, as well as chronic obstructive pulmonary disease, sleep apnea, and heart failure can activate neuroimmune markers such as interleukin (IL)-1β in the brain and cause social behavioral impairment [28]. In addition, an animal study reported increased plasma tumor necrosis factor (TNF)-α and IL-6 levels in mice exposed to 5% oxygen for one hour, and a human study also indicated increased plasma IL-6 levels at high altitudes [29]. Moreover, the blood levels of inflammatory markers and oxidative stress markers are known to be increased in persons with sleep apnea [30].

Exposure to endocrine-active metals, such as lead and mercury, during the brain developmental period can alter the hypothalamic-pituitary-adrenal (HPA) axis and neurobehaviors [31,32,33]. In a previous study, we showed that high-dose phthalate di-(2-ethylhexyl) phthalate (DEHP) exposure during adolescence induced neuroinflammation via neuroimmune biomarkers in the hypothalami of allergic asthma mice [34]. A recent study indicated that prenatal stress enhanced the neurotoxicity of lead and mercury by simultaneously affecting the HPA axis and central nervous system (CNS), including the hippocampus and the mesocorticolimbic system [35].

Taken together, we hypothesize that asthma can cause brain hypoxia and elicit inflammatory responses via inducing inflammatory and oxidative stress markers; asthma also may induce cognitive impairment via modulation of the hippocampal memory function-related genes. However, the effects of BPA exposure on the cognitive function and neuroinflammatory responses in asthmatic subjects are largely unknown, as seen in Figure 1. Therefore, we investigated the effects of BPA exposure on (1) the cognitive function using the novel object recognition ability test and the hippocampal NMDA receptor expressions, and (2) neuroinflammation using inflammatory biomarkers such as IL-1β, TNF-α, cyclooxygenase-2 (COX2), ionized calcium binding adapter molecule (Iba)1, and behavior-related genes, such as the estrogen receptor (ER)α and oxytocin receptor (oxtr), in the hypothalamus of a mouse model with allergic asthma (AA).

## 2. Materials and Methods

### 2.1. Animals

Five-week-old male C3H/HeJJcl mice were purchased from Japan Clea Co. (Tokyo, Japan), and the animals were housed in cages under controlled environmental conditions (temperature 22–26 °C; humidity, 40–69%; lights on 07:00–19:00 h). The mice were fed a commercial diet (CE-2; Japan Clea) and given water ad libitum. This study was approved by the Animal Care and Use Committee of the National Institute for Environmental Studies, Japan.

### 2.2. Experimental Design

The three doses of BPA were used in the present study as described previously [20]. These doses were equivalent to 0.25, 5, and 100 times the estimated peak exposure from atmosphere in Japan (0.0003 μg/kg/day) [36]. Exposure doses were calculated based on the body weight of the mice. For the intratracheal instillation model, we needed to give anesthesia in every exposure; to exclude the effect of anesthesia, we reduced the daily exposure to anesthesia to weekly exposure. The predicted daily maximum exposure of BPA in indoor air is 0.0003 μg/kg/day (Ministry of the Environment, Japan), set by converting this to the amount of exposure per week. A mouse model of AA was created by repeatedly administration of ovalbumin (OVA) intratracheally. Six-week-old male mice were divided into eight experimental groups: (1) Vehicle; (2) 0.0625 pmol BPA/animal 25 g/week (BPA-L); (3) 1.25 pmol BPA/animal 25 g/week (BPA-M); (4) 25 pmol BPA/animal 25 g/week (BPA-H); (5) OVA; (6) OVA + BPA-L; (7) OVA + BPA-M; and (8) OVA + BPA-H. Mice were anesthetized with 4% halothane (Takeda Chemical Industries, Ltd., Osaka, Japan) and intratracheally instilled with 100 μL aliquots of an aqueous suspension. The vehicle group received phosphate buffered saline (PBS, pH 7.4; Thermo Fisher Scientific, Inc., Chicago, IL, USA) containing 0.0005% ethanol (Nacalai Tesque, Inc., Kyoto, Japan) once per week for six weeks; the OVA group received 1 μg of OVA (10 μg/mL; Sigma-Aldrich Co., St. Louis, MO, USA) dissolved in PBS containing 0.0005% ethanol every two weeks for a period of six weeks; the BPA groups received 0.0625, 1.25, or 25 pmol of BPA (Sigma-Aldrich Co., St. Louis, MO, USA) in PBS containing 0.0005% ethanol once per week for six weeks; and the OVA + BPA groups received a combined administration of BPA and OVA. At 11 weeks of age, a novel object recognition test was conducted before the final administration of OVA, hypothalamus was collected later, and neuroimmune and endocrine biomarkers were examined using real-time reverse transcription polymerase chain reaction (RT-PCR) method.

### 2.3. Novel Object Recognition Test

The novel object recognition test was first introduced by Ennaceur and Delacour [37] to assess the ability of rats to recognize a new object from a familiar one. We performed a novel object recognition test over a period of four days including a habituation phase (15 min/day for two consecutive days), a training phase (10 min for one day), and a test phase (5 min for one day) in each mouse at 11 weeks of age, as described previously [38]. During the habituation phase, the mouse was placed in a rectangular cage (50 × 50 × 40 cm) made of acryl for 15 min per day for two days without an object. During the training phase, two identical objects (6 × 7 × 8 cm) were placed near the corners on one wall of the rectangular cage (10 cm from each adjacent wall). The mouse was placed into the center of the cage facing the opposite wall and was allowed to explore both objects for 10 min. Exploration was defined as the mouse pointing its nose toward the object at a distance of less than 2 cm. We did not record the time spent sitting or resting against the object. Twenty-four hours after the training phase, during the test phase, one of the old objects was replaced with a novel object (8 × 9 × 10 cm) and was presented to each mouse for 5 min. To control for odor cues, the open field arena and the objects were thoroughly cleaned with water, dried, and ventilated for a few minutes between mice. The object exploration time was recorded using a video assisted tracking system (Muromachi Kikai Co. Ltd., Tokyo, Japan). Discrimination between the two objects was calculated using a discrimination index (DI) as follows: DI = ([novel object exploration time/total exploration time] − [familial object exploration time/total exploration time]) × 100. This equation takes into account individual differences in the total amount of exploration time [39]. We also performed object preference test by the control and BPA-treated mice with OVA immunization and or without OVA immunization in BPA-H group. The positions of the objects in the test and the objects used as novel or familiar were counterbalanced between the mice.

### 2.4. Quantification of mRNA Expression Levels

At 11 weeks of age, the mice (*n* = 5~6 from each group) were sacrificed under deep pentobarbital anesthesia and the hippocampus and hypothalamus were collected from each group of mice and frozen quickly in liquid nitrogen, then stored at −80 °C until the extraction of the total RNA. Briefly, the total RNA was extracted from the hippocampal samples using the BioRobot EZ-1 and EZ-1 RNA tissue mini kits (Qiagen GmbH, Hilden, Germany). Then, the purity of the total RNA was examined, and the quantity was estimated using the ND-1000 NanoDrop RNA Assay protocol (NanoDrop, Wilmington, DE, USA), as described previously [40]. Next, we performed first-strand cDNA synthesis from the total RNA using SuperScript RNase H-Reverse Transcriptase II (Invitrogen, Carlsbad, CA, USA), according to the manufacturer’s protocol. We examined the hippocampal mRNA expression levels using a quantitative real-time RT-PCR method and the Applied Biosystems (ABI) Prism 7000 Sequence Detection System (Applied Biosystems Inc., Foster City, CA, USA). The tissue 18S rRNA level was used as an internal control. The primer sequences used in the present study are shown below. Some primers IL-1β, NM_008361; COX2, NM_011198; Iba1, NM_019467; ERα, NM_007956; oxtr, NM_001081147; NR1, NM_008169; NR2A, NM_008170; NR2B, NM_008171 were purchased from Qiagen, Sample and Assay Technologies. Other primers were designed in our laboratory as follows: 18S (forward 5′-TACCACATCCAAAAGGCAG-3′, reverse 5′-TGCCCTCCAATGGATCCTC-3′), and TNF-α (forward 5′-GGTTCCTTTGTGGCACTTG-3′, reverse 5′-TTCTCTTGGTGACCGGGAG-3′). Data were analyzed using the comparative threshold cycle method. Then, the relative mRNA expression levels were expressed as mRNA signals per unit of 18S rRNA expression.

### 2.5. Immunohistochemical Analyses

Microglial activation in the hippocampus was examined in BPA-H groups with or without OVA. The hippocampal tissue sections were stained with microglial marker Iba1 as described previously [41]. Briefly, the brain sections were immersed in absolute ethanol, followed by 10% H_2_O_2_ for 10 min each at room temperature. After rinsing in 0.01-M phosphate buffer saline, the sections were blocked with 2% normal swine serum in PBS for 30 min at room temperature and then reacted with goat polyclonal anti-Iba1 (diluted 1:100; abcam: ab5076; Tokyo, Japan) in PBS for 1 h at 37 °C. Thereafter, the sections were reacted with biotinylated donkey anti-rabbit IgG (1:300 Histofine; Nichirei Bioscience, Tokyo, Japan) in PBS for 1 h at 37 °C. The sections were then incubated with peroxidase-tagged streptavidin (1:300, ABC KIT) containing PBS for 1 h at room temperature. After a further rinse in PBS, Iba1 immunoreactivity was detected using a Dako DAB Plus Liquid System (Dako Corp., Carpinteria, CA, USA). To detect the immunoreactivity of Iba1 in the hippocampus, photomicrographic digital images (150 dpi, 256 scales) of the hippocampal regions were taken using a charged coupled device (CCD) camera connected to a light microscope.

### 2.6. Statistical Analysis

The statistical analyses were performed using the Statcel4 statistical analysis system for Microsoft Excel, Version 4.0 (OMS Publishing Inc., Tokyo, Japan). Regarding the novel object recognition test, object exploration time was analyzed by a non-parametric Mann–Whitney U test, and DI was analyzed by non-parametric multiple comparison Steel–Dwass test. Body weight and brain weight were analyzed by multiple comparison Steel–Dwass test within OVA (−) or OVA (+) groups and non-parametric Mann–Whitney U test between OVA (−) and OVA (+) groups. Messenger RNA expressions were analyzed by non-parametric multiple comparison Steel–Dwass test. Differences were considered significant at *p* < 0.05.

## 3. Results

### 3.1. Body Weight and Brain Weight

To determine the general toxicity of BPA and OVA co-exposure, we measured the body and brain weights of the male mice after the completion of the exposure periods at the time of sampling. No significant difference of the body weight or the brain weight was observed among the exposure groups and between OVA (−) and OVA (+) groups, as seen in Table 1.

### 3.2. Effect of Bisphenol A (BPA) Exposure on Novel Object Recognition Test

Firstly, we performed the novel object recognition test in OVA-immunized AA mouse models. We used five groups including vehicle, OVA, OVA + BPA-L, OVA + BPA-M, and OVA + BPA-H. As a result, OVA alone, OVA + BPA-L, and OVA+BPA-M groups showed no different exploration time between familial and novel objects whereas OVA+BPA-H exposed male mice showed a significant decreased exploration time to novel object when compared to the familiar object, as seen in Figure 2A (*p* < 0.05). Discrimination between familiar and novel objects was calculated using a DI. In addition, DI in BPA-H exposed mice was significantly reduced when compared with that in the vehicle -exposed mice, as seen in Figure 2B (*p* < 0.05). The exploration time to the familial object in OVA + BPA-H group is higher than vehicle group, conversely, the exploration time to the novel object in OVA + BPA-H group is lower than vehicle group (*p* = 0.0249). This result may be due to that the vehicle group expressed more interest in the novel object than the familial object (*p* = 0.0104) and OVA + BPA-H group expressed little interest in the novel object in comparison with the familial object. These findings indicate that the mice exposed to OVA + BPA-H could not discriminate a novel object from a familiar object on no interest in a new object. In addition, OVA alone, OVA + BPA-L, and OVA + BPA-M groups could not discriminate novel from familial object. These findings prompted us to investigate whether those effects were due to OVA or BPA or OVA + BPA co-exposure. Thus, we conducted the next experiment of novel object recognition using four groups including vehicle, BPA-H alone, OVA alone, and OVA+BPA-H, as seen in Figure 3.

According to our results, vehicle and BPA-H groups showed significantly increased exploration time (*p* = 0.0274, *p* = 0.0104, respectively) to novel object but not in OVA alone and OVA + BPA-H groups, as seen in Figure 3B. Our two experiments showed moderately different results, as seen in Figure 2A and Figure 3A as well as in Figure 2B and Figure 3B. Because of the separate nature of the experiment, environmental conditions such as temperature, humidity, and light may have influenced animal behavior. Vehicle group and OVA alone group showed similar results in two experiments. Although different results of OVA + BPA-H between two experiments were observed, these two results showed no interest of novel object in Figure 2A and poor discrimination between familial and novel objects in Figure 3A. Taken together, our findings indicate that BPA-H alone could not affect memory function in normal condition and could affect in individuals with underlying asthma or allergy.

### 3.3. Effect of BPA Exposure on the mRNA Expressions of N-methyl-D-aspartate (NMDA) Receptor Subunits in the Hippocampus

The hippocampal NMDA receptors play an important role in the object recognition memory [42]. The mRNA expression levels of the NMDA receptor subunits NR1, NR2A, and NR2B in the hippocampus were investigated. Significantly reduced mRNA expression level of the NMDA receptor subunits NR2B was found in the hippocampi of mice exposed to OVA+BPA-H when compared with the control, as seen in Figure 4C (*p* < 0.05).

### 3.4. Effect of BPA Exposure on the mRNA Expressions of Inflammatory Cytokines in the Hypothalamus

To detect the BPA-induced inflammation in the hypothalamus, we investigated the effect BPA exposure on inflammatory markers such as IL-1β and TNF-α in the hypothalamus of mice with or without OVA immunization. No remarkable changes of IL-1 β and TNF-α mRNA expression levels were observed within and between groups with or without OVA immunization, as seen in Figure 5.

### 3.5. Effect of BPA Exposure on the mRNA Expressions of Inflammatory and Microglia Markers in the Hypothalamus

Furthermore, the mRNA expression of potent inflammatory marker COX2 and Iba1, a microglial marker, in the hypothalamus were also detected. There were no significant differences in COX2 and Iba1 mRNA expression levels within and between groups with or without OVA immunization, as seen in Figure 6.

### 3.6. Effect of BPA Exposure on the mRNA Expressions of Estrogen Receptor and Oxytocin Receptor in the Hypothalamus

BPA has estrogenic and anti-androgenic effects on physiological and behavioral functions [1,2]. ERα and ERβ are expressed in the brain of both male and female rodents. ERα expression in the hypothalamus contributes to learning and memory functions [43]. We have investigated the effect BPA exposure on ERα and oxtr expression in the hypothalamus of mice with or without OVA immunization. We found that the expression levels of ERα was not different between control and BPA exposed groups with or without OVA immunization. However, oxtr mRNA expression was increased significantly in BPA-M group compared to vehicle in non-OVA-exposed mice (*p* = 0.037, Figure 7B). In addition, oxtr mRNA expression was increased significantly in BPA-M group compared to vehicle group, as seen in Figure 7B, and OVA alone exposure group when compared with vehicle only group (*p* = 0.006), as seen in Figure 7D.

### 3.7. Immunohistochemical Analyses

Hippocampal microglial activation was not different between vehicle group and BPA-H alone, OVA alone, and OVA + BPA-H-exposed mice. Representative digital photomicrographs of Iba1-immunostained sections taken from the hippocampus of the control and exposure groups are shown in Figure 8.

## 4. Discussion

This study investigated the effects of exposure to BPA in susceptible groups, such as subjects with allergic airway inflammation, and examined the effects of co-exposure to BPA and an allergen (OVA) in adolescence. In regard to AA mouse models, we have already reported that airway inflammation and increased serum levels of OVA-specific antibodies were observed in the same OVA-immunized mouse models [20]. The major findings of this study were the appearance of impaired novel object recognition ability in the male allergic asthmatic mice following high-dose BPA exposure, accompanied by downregulation of the NMDA receptor subunit NR2B in the hippocampus. The mRNA expressions of inflammatory markers, such as IL-1, TNF-α, COX2, and microglia marker Iba1, were not different in the hypothalami of the allergic asthmatic mice exposed to BPA. Moreover, the expression levels of ERα and oxtr were not different between the control and BPA-exposed groups, regardless of the status of OVA immunization. To the best of our knowledge, this is the first study to show that intratracheal instillation of high-dose BPA alone could not affect memory function in normal condition and could affect in individuals with underlying asthma or allergy.

In humans, a common pathway of exposure to BPA is via dietary ingestion. However, other routes of exposure, such as inhalation, cannot be excluded because BPA has been identified in common house dust and has been found in the atmosphere [20]. Three doses of BPA such as BPA-L, BPA-M and BPA-H- were used in that study which were equivalent to 0.25, 5, and 100 times the estimated peak exposure to BPA from the atmosphere in Japan (0.0003 μg/kg/day) [36]. We selected the intratracheal instillation route to detect the effects of specific airway exposure to BPA. Human studies have indicated the existence of an association between BPA exposure and behavioral problems such as anxiety and depression in a gender-dependent manner in children [44,45,46]. In animal studies, BPA-induced working memory deficits in the adult male monkey have been reported [47]. BPA can enter the human body via ingestion, inhalation, and transdermal pathways. We examined the effects of intratracheal instillation of BPA on the novel object recognition ability in male mice and found that male allergic asthmatic mice exposed to high-dose BPA showed appearance of impaired novel object recognition ability.

The nervous and immune systems communicate with each other, and neuroimmune interactions have a role in the pathophysiology of neurodegenerative diseases. Immune responses in the CNS involve not only the activation of the resident cells such as microglia and astrocytes but also infiltration of circulating immune cells, such as monocytes, neutrophils, and T cells. Both activated resident cells and infiltrating cells in the CNS act as regulators of immunity and modulators of the neuronal and glial functions [48]. Interactions among neurons, immune cells, and neurotrophins are potentially responsible for the regulation and control of neuroimmune crosstalk. Little was known regarding the effects of BPA exposure on the neuroimmune biomarker expression levels in the hypothalamus of allergic asthmatic mice. Therefore, to evaluate the neuroimmune interactions in animals exposed to BPA, we examined potential neuroimmune biomarker levels, such as the expression levels of proinflammatory cytokines and an oxidative stress marker, as well as the levels of microglia markers in allergic asthmatic mice.

Juvenile age is the last developmental stage of the central nervous system [49] and there are limited studies for this age group. In the present study, we developed an AA mouse model using OVA immunization and investigated the effects of exposure to BPA on the memory function and expression levels of neuroimmune biomarkers in the hypothalami of juvenile mice. The hippocampus plays a partial role in recognition memory [50]. Our previous studies also showed the involvement of the hippocampal NMDA receptor in novel object recognition memory [38,40]. Thus, the effects of BPA exposure on novel object recognition memory were examined in AA mouse models and it was found that the BPA-H-exposed mice showed poor discrimination between familiar and novel objects. Second, to confirm these effects were due to BPA or OVA or co-exposure, the object recognition test was performed in mice exposed to vehicle, BPA alone, OVA alone, and OVA + BPA-H. OVA alone and OVA + BPA-H-exposed mice showed poor discrimination between novel and familial objects. Our findings suggest that a high dose of BPA impaired memory function in allergic asthmatic individuals; however, it did not affect memory function in normal individuals.

In contrast, no difference in the effects on microglia activation was observed between the vehicle and test compound exposure groups. The hypothalamus is characterized as the principal brain region for neuroendocrine activity: it sends signals to and receives information from endocrine glands to maintain body homeostatic functions. Moreover, the hypothalamus also regulates body processes such as metabolism, immunity reactions, aggression, emotions, learning, and memory [51,52]. Accordingly, we selected the hypothalamus to detect the roles of neuroimmune biomarkers in BPA-induced neurotoxicity in AA mouse models.

Cytokines are the principal biomarkers of neuroinflammation in neuropathology and neurodegenerative processes [53,54]. Previously, we demonstrated that proinflammatory cytokine mRNA levels were upregulated in the brains of mice exposed to environmental pollutants, such as carbon black nanoparticles, nanoparticle-rich diesel exhaust, and volatile organic compound toluene [55,56,57,58,59]. COX is required for the conversion of arachidonic acid to prostaglandins, which are involved in the inflammatory responses in the brain. Exposure to BPA and allergen from 5 to 11 weeks old may induce chronic stress. Chronic stressors disrupt cytokine homeostasis through the activation of the HPA axis, sympathetic adrenal medullary axis, and vagal fibers, promoting the secretions of glucocorticoids, catecholamines, and acetylcholine to inhibit proinflammatory cytokine secretion [60]. However, no significant changes of mRNA expression levels of inflammatory markers in the hypothalamus were observed in this study.

Microglia are the major immune cells resident in the brain [61] and the principal source of brain immune mediators. Activated microglia generally release several cytotoxic substances, including free radicals, excitatory amino acids, arachidonic acid derivatives, and cytokines. We detected major microglia activation using the microglia marker Iba1 but no significant difference in the Iba1 mRNA level in the hypothalamus between the BPA-exposed mice, regardless of the status of OVA immunization and the control mice.

Estrogen receptors show a sex-specific distribution in the rat hypothalamus [62]. BPA exerts potential neurotoxic effects on the neuronal morphology and brain functions by binding ERs in a different manner. In the present study, we examined the effects of juvenile BPA exposure on the expression of ERα, which has been implicated in synaptic plasticity and in the pathophysiology of anxiety behaviors [63]. We found no difference in the ERα expression level between the BPA- exposed mice that either received or did not receive OVA immunization. This indicates that juvenile BPA exposure may not affect the hypothalamic ER in the regulation of cognitive functions. Brain oxytocin plays a role in sociosexual behaviors [64,65] and anxiolytic effects [66]. Disrupted ontogeny of the oxytocin signaling pathways may underlie juvenile affective behavior. Prenatal BPA exposure resulted in significantly lower whole brain levels of oxytocin in mice just prior to birth when compared with controls [67]. In this study, BPA-M-exposed mice showed significantly increased ortr mRNA expression level when compared with vehicle-exposed mice. We did not know exactly the cause of this effect, but it was suggested that the dose-specific effect of BPA alone exposure may affect the oxytocin receptor expression in the hypothalamus.

From our findings, co-exposure to allergen plus high-dose BPA from the juvenile period to adulthood may affect novel object recognition ability, accompanied by alterations in memory function-related gene expressions in the hippocampus. In the present study, significant responses were observed in the animals exposed to high-dose BPA. Previously, we have shown that intratracheal administration of high-dose DEHP during adolescence induced neuroinflammation by modulating the neuroimmune biomarker expression levels in the hypothalamus in AA mouse models [34]. One possibility is that a very high level of BPA exposure is needed for the expression levels of the neuroimmune biomarkers to be affected. Another possibility is that the duration of exposure, route of administration, and exposure schedule also influence the dose–response effects. The limitations of our study were the small sample size of animals and lack of behavioral assessment in BPA-L, BPA-M-exposed groups. In a future study, we will increase the number of animals and perform behavioral tests for different doses of BPA-exposed mice.

## 5. Conclusions

Appearance of impairment of the novel object recognition ability was observed in the high-dose BPA-exposed allergic asthmatic mice. In addition, the allergic asthmatic mice also showed downregulation of neurological biomarkers, such as NMDA receptor subunit NR2B in the hippocampus but no significant effect on immunological biomarkers in the hypothalamus. This is the first study to show that, in the presence of allergens, exposure to high-dose BPA may affect memory by modulating the memory function-related genes in the hippocampus.

## Figures and Tables

**Figure 1 ijerph-16-03770-f001:**
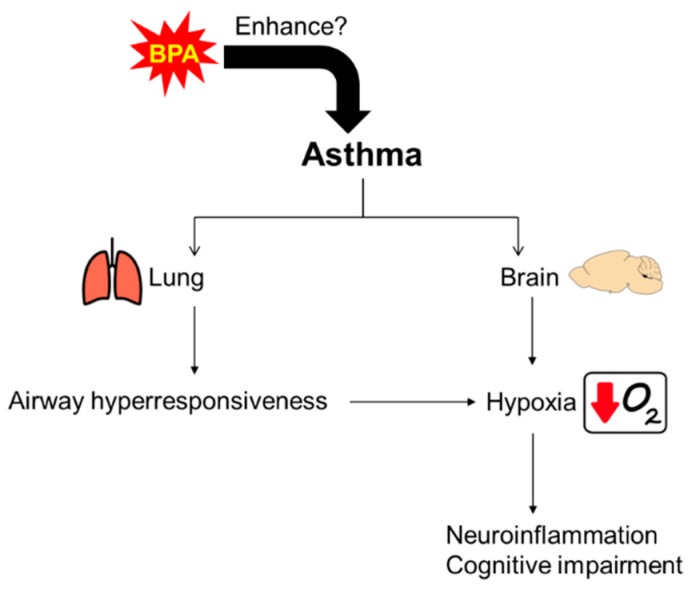
Illustration of possible brain–lung network in asthma. Asthma may induce brain hypoxia and cause inflammatory responses via inflammatory and oxidative stress markers and also induce cognitive impairment via hippocampal memory function related genes. Bisphenol A (BPA) exposure may enhance these effects.

**Figure 2 ijerph-16-03770-f002:**
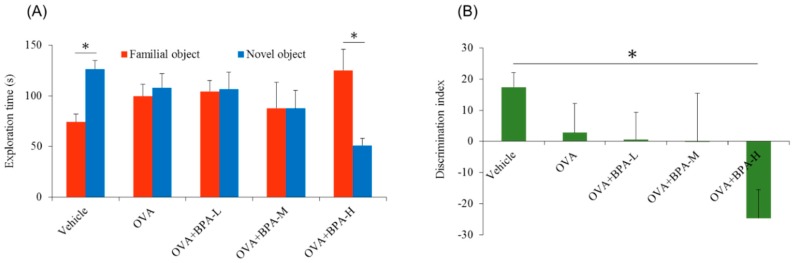
Novel object recognition test. (**A**) Test phase showing exploration time to familial or novel object, and (**B**) discrimination ability between familial and novel object. Each bar represents the mean + SE (*n* = 5~6, * *p* < 0.05).

**Figure 3 ijerph-16-03770-f003:**
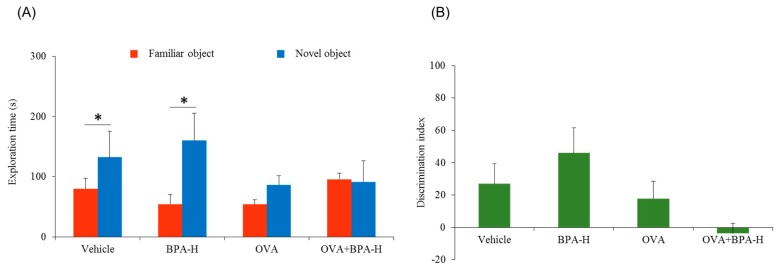
Novel object recognition test. (**A**) Test phase showing exploration time to familial or novel object, and (**B**) discrimination ability between familial and novel object of BPA-H exposed mice with or without ovalbumin (OVA). Each bar represents the mean + SE (*n* = 5~6, * *p* < 0.05).

**Figure 4 ijerph-16-03770-f004:**
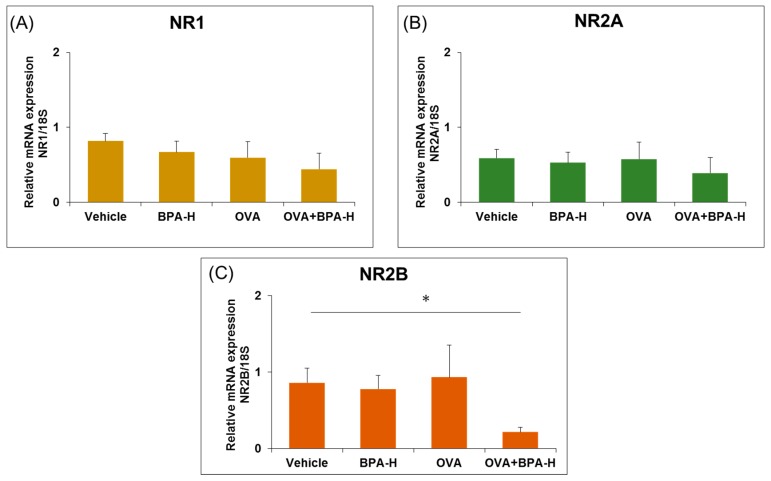
Expression level of N-methyl-D-aspartate (NMDA) receptor subunits (**A**) NR1, (**B**) NR2A and (**C**) NR2B in the hippocampus of BPA-H group with or without OVA. Each bar represents the mean + SE (*n* = 5~6, * *p* < 0.05 versus vehicle).

**Figure 5 ijerph-16-03770-f005:**
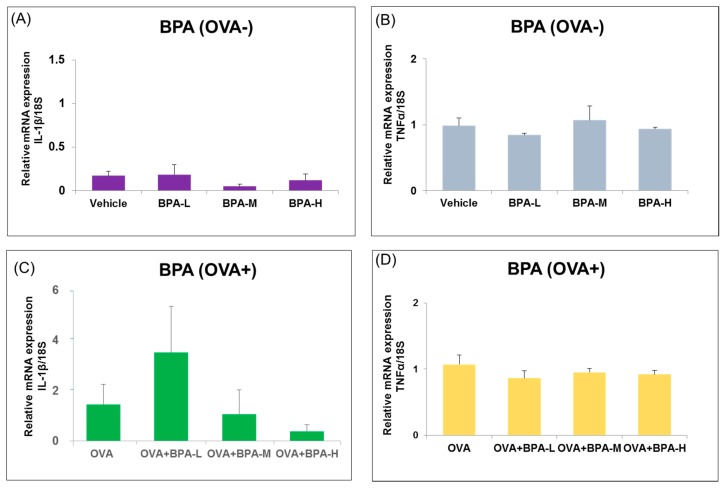
Expression level of inflammatory cytokines (IL-1β and TNF-α) in the hypothalamus of male mice exposed t to BPA without OVA (**A**,**B**) and with OVA (**C**,**D**). Each bar represents the mean + SE (*n* = 5~6).

**Figure 6 ijerph-16-03770-f006:**
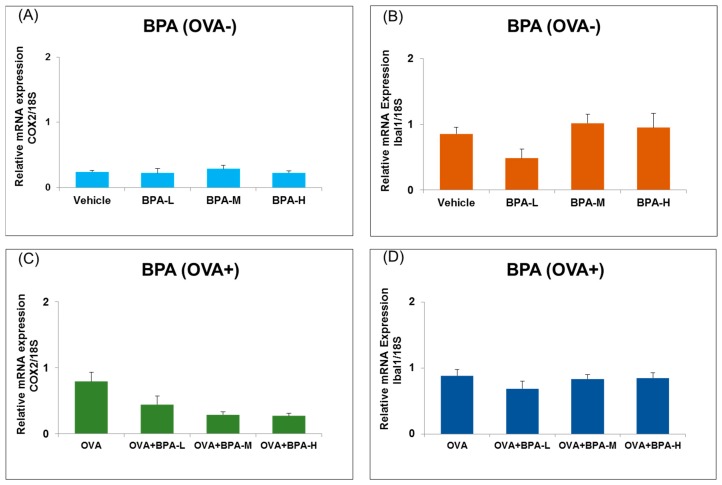
Expression level of inflammatory markers (COX2) and microglia marker (Iba1) in the hypothalamus of male mice exposed to BPA without OVA (**A**,**B**) and with OVA (**C**,**D**). Each bar represents the mean + SE (*n* = 5~6).

**Figure 7 ijerph-16-03770-f007:**
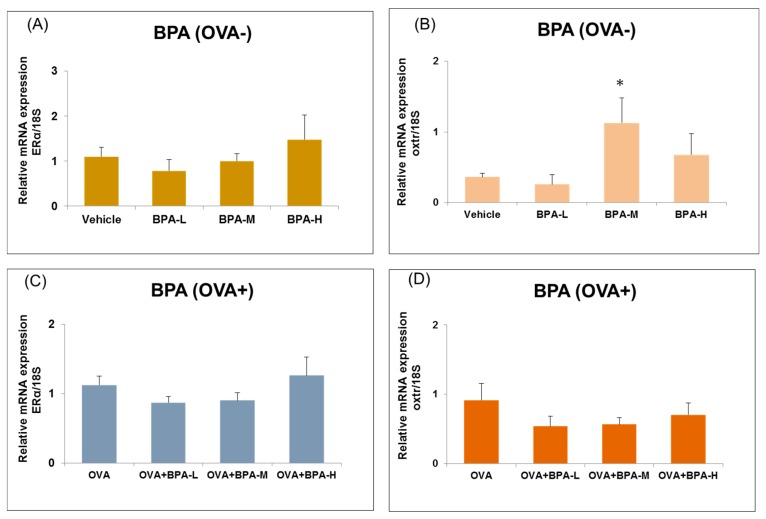
Expression level of estrogen receptor (ERα) and oxytocin receptor (oxtr) in the hypothalamus of male mice exposed to BPA without OVA (**A**,**B**) and with OVA (**C**,**D**). Each bar represents the mean + SE (*n* = 5~6, * *p* < 0.05 versus vehicle).

**Figure 8 ijerph-16-03770-f008:**
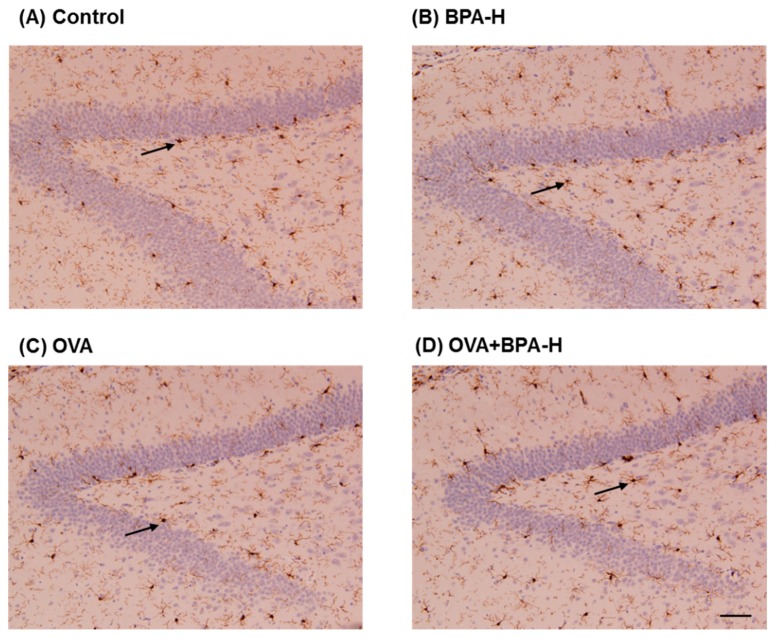
Representative photomicrographs showing the DG (dentate gyrus) area of the hippocampi immunostained with Iba1 in (**A**) control, (**B**) BPA-H, (**C**) OVA and (**D**) OVA+BPA-H exposed mice. Black arrows indicate activated microglia. Scale bar = 50 µm.

**Table 1 ijerph-16-03770-t001:** Body weight and brain weight of eight experimental groups.

Exposure Groups	Body Weight (g)	Brain Weight (mg)
Vehicle	27.78 ± 0.59	450.30 ± 5.83
BPA-L	27.47 ± 1.30	436.77 ± 12.65
BPA-M	28.98 ± 0.57	426.82 ± 4.86
BPA-H	28.33 ± 0.30	434.38 ± 2.69
OVA	29.32 ± 0.71	441.55 ± 4.28
OVA + BPA-L	27.54 ± 0.79	450.84 ± 5.44
OVA + BPA-M	29.22 ± 0.78	438.76 ± 6.00
OVA + BPA-H	28.34 ± 0.48	440.00 ± 6.67

Each data represents the mean ± SE (*n* = 5~6).

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
