# Peer review of "Memory Function, Neurological, and Immunological Biomarkers in Allergic Asthmatic Mice Intratracheally Exposed to Bisphenol A"

_ijerph, 2019, doi:10.3390/ijerph16193770_

Round 1

Reviewer 1 Report

I appreciate the efforts of the authors to ameliorate the manuscript that, in my opinion, needs some further improvements, including English editing.

Methods

Page 3, paragraph line 30. Consider to shift this paragraph from methods to 3.2 section of Results.

Page 3, line 38. “..the next experiment” instead “next experiment”.

Results

I understand that results reported in Figure 3 refer to a further experiment, but in this second experiment exploration times for familial and novel objects in OVA + BPA-H group are considerably different compared to those reported in the first experiment (as reported in Figure 2), and, differently from Figure 2, in Figure 3B, DI does not significantly differ from DI of the vehicle. The authors should explain the reason of such differences. Is the novel object used different in the two experiments? Are different the experimental conditions (e.g., light, temperature, etc.)? On the other hand, results for vehicle and OVA groups are similar in the two experiments.

Page 6, line 9. Consider to delete “and” before “….but not in OVA alone and OVA+BPA‑H”.

Page 9, lines 14-15. The verb is missing.

Discussion

Page 11, lines 38-41. The authors could consider not to copy the same paragraph written in Results (page 6, lines 8-11) to Discussion.

Page 12, line 5. Why did the authors state that there is a downregulation of inflammatory markers in the study? In the results they reported any significant difference in expression of these markers within and between groups with or without OVA immunization.

Page 12, lines 27-28. This statement is not completely correct, given the statistically significant increase of oxtr in BPA-M group.

Unfortunately, I have noted that the authors maintained an excessive use of “in the present study”, despite my previous comment. In addition, instead of “route”, sometimes the term “pathway” could be used (page 11, paragraph 3-15).

Author Response

Responses to the Reviewer 1

We really appreciate the Reviewer’s invaluable comments and suggestions.

Comments and Suggestions for Authors

I appreciate the efforts of the authors to ameliorate the manuscript that, in my opinion, needs some further improvements, including English editing.

Methods

Page 3, paragraph line 30. Consider to shift this paragraph from methods to 3.2 section of Results.

   As suggested by the Reviewer, we shifted this paragraph from Methods to 3.2 section of Results in our revised manuscript.

Page 3, line 38. “..the next experiment” instead “next experiment”.

   As commented by the Reviewer, we changed “next experiment” to “..the next experiment” in our revised manuscript.

Results

I understand that results reported in Figure 3 refer to a further experiment, but in this second experiment exploration times for familial and novel objects in OVA + BPA-H group are considerably different compared to those reported in the first experiment (as reported in Figure 2), and, differently from Figure 2, in Figure 3B, DI does not significantly differ from DI of the vehicle. The authors should explain the reason of such differences. Is the novel object used different in the two experiments? Are different the experimental conditions (e.g., light, temperature, etc.)? On the other hand, results for vehicle and OVA groups are similar in the two experiments.

   As commented by the Reviewer, our two experiments showed some different results (Figure 2A and 3A, and Figure 2B and 3B). Because of the separate experiment, the environmental conditions like temperature, humidity and light may influence the animal behavior. Vehicle group and OVA alone group showed similar results in two experiments. Although different results of OVA+BPA-H between two experiments were observed, these two results showed no interest of novel object in Figure 2A and poor discrimination between familial and novel objects in Figure 3A.

  We provided this issue in our revised manuscript.

Page 6, line 9. Consider to delete “and” before “….but not in OVA alone and OVA+BPA‑H”.

   As suggested by the Reviewer, we have deleted “and” before “….but not in OVA alone and OVA+BPA‑H” in our revised manuscript.

Page 9, lines 14-15. The verb is missing.

   As commented by the Reviewer, we added verb in that sentence as follows;

No remarkable changes of IL-1 b and TNF-a mRNA expression levels were observed within and between groups with or without OVA immunization

Discussion

Page 11, lines 38-41. The authors could consider not to copy the same paragraph written in Results (page 6, lines 8-11) to Discussion.

    As commented by the Reviewer, we revised our sentences as follows in our revised manuscript;

    Next step, to confirm these effects are due to BPA or OVA or co-exposure, the object recognition test was performed in mice exposed to vehicle, BPA alone, OVA alone and OVA+BPA-H. OVA alone and OVA+BPA-H exposed mice showed poor discrimination between novel and familial objects. Our findings suggest that high dose of BPA alone could not influence memory function and could influence in AA groups.

Page 12, line 5. Why did the authors state that there is a downregulation of inflammatory markers in the study? In the results they reported any significant difference in expression of these markers within and between groups with or without OVA immunization.

    We deleted that sentence in our revised manuscript.    

Downregulation of the inflammatory markers in the hypothalamus in the present study could be attributable to the animals being in the early stage of chronic stress.

Page 12, lines 27-28. This statement is not completely correct, given the statistically significant increase of oxtr in BPA-M group.

  As commented by the Reviewer, we revised our sentence as follows in our revised manuscript;

In the present study, BPA-M-exposed mice showed significantly increased ortr mRNA expression level compared to vehicle-exposed mice in this study. We did not know exactly the cause of this effect, but it was suggested that dose-specific effect of BPA alone exposure may affect the oxytocin receptor expression in the hypothalamus.

Unfortunately, I have noted that the authors maintained an excessive use of “in the present study”, despite my previous comment. In addition, instead of “route”, sometimes the term “pathway” could be used (page 11, paragraph 3-15).

   As suggested by the Reviewer, we revised our sentences in that paragraph as follows in our revised manuscript;

 In humans, a common pathway of exposure to BPA is via dietary ingestion. However, other routes of exposure, such as inhalation, cannot be excluded, because BPA has been identified in indoor house dust and in the atmosphere [20]. In the present study, Three doses of BPA such as BPA-L, BPA-M and BPA-H- were used in that study which were equivalent to 0.25, 5, and 100 times the estimated peak exposure to BPA from the atmosphere in Japan (0.0003 μg/kg/day) [36]. We selected the intratracheal instillation route in this study to detect the effects of specific airway exposure to BPA. Human studies have indicated the existence of an association between BPA exposure and behavioral problems such as anxiety and depression in a gender-dependent manner in children [44-46]. In animal studies, it has been reported that BPA induced working memory deficits in the adult male monkey [47]. BPA can enter the human body via the ingestion, inhalation and transdermal pathways. In the present study, We examined the effects of intratracheal instillation of BPA on the novel object recognition ability in male mice and found that male AA mice exposed to high-dose BPA showed appearance of impaired novel object recognition ability.

Reviewer 2 Report

Please highlight the limitations of this study. Why did the BPA groups receive BPA once per week for 6 weeks, because BPA exposure is usually a continuous process. 

Author Response

Responses to the Reviewer 2

Please highlight the limitations of this study. Why did the BPA groups receive BPA once per week for 6 weeks, because BPA exposure is usually a continuous process.

   As suggested by the Reviewer, we provided limitation our study in Discussion section in our revised manuscript as follows;

   The limitations of our study are small sample size of animal, lack of behavioral assessment in BPA-L, BPA-M-exposed groups. In our future study, we increase the number of animals, perform behavioral test in different doses of BPA-exposed mice.

We agree that BPA exposure is usually a continuous process. However, for intratracheal instillation model, we need to give anesthesia in every exposure and to exclude the effect  of anesthesia, we reduce the anesthesia daily exposure to weekly exposure. The predicted daily maximum exposure of BPA in indoor air is 0.0003μg/kg /day (Ministry of the Environment, Japan) and set it by converting to the amount of exposure per week.

  We provided this issue in 2.2. Experimental design our revised manuscript.

Reviewer 3 Report

I don't know why the author uses the C3H model as an asthma model since better models have been reported

Exp Anim. 2014; 63(4): 435–445.   “In conclusion, BALB/c and NC/Nga mice demonstrated markedly increased IgE reactions. Inflammatory cell counts in BALF were increased in the treated groups of all strains, especially BALB/c, NC/Nga, and CBA/J strains. Cytokine levels in LNs were increased in all treated groups except for C3H/HeN and were particularly high in BALB/c and NC/Nga mice. According to our results, we suggest that BALB/c and NC/Nga are highly susceptible to respiratory allergic responses and therefore are good candidates for use in our model for detecting environmental chemical respiratory allergens.”

On the other hand, C3H mice lack TLR4 and it has been reported that this TLR is important for memory. (2012) Evidence for a Developmental Role for TLR4 in Learning and Memory. PLoS ONE 7(10): e47522. This paper concludes:  “Our findings suggest that TLR4 has a developmental role in shaping spatial and contextual learning and memory.”

How does the author show that the memory problems he observes are not due to the absence of TLR4?

The discussion is very long and does not really discuss the results.

Author Response

Responses to the Reviewer 3

We really appreciate the Reviewer's invaluable comments and suggestions.

I don't know why the author uses the C3H model as an asthma model since better models have been reported

Exp Anim. 2014; 63(4): 435–445.   “In conclusion, BALB/c and NC/Nga mice demonstrated markedly increased IgE reactions. Inflammatory cell counts in BALF were increased in the treated groups of all strains, especially BALB/c, NC/Nga, and CBA/J strains. Cytokine levels in LNs were increased in all treated groups except for C3H/HeN and were particularly high in BALB/c and NC/Nga mice. According to our results, we suggest that BALB/c and NC/Nga are highly susceptible to respiratory allergic responses and therefore are good candidates for use in our model for detecting environmental chemical respiratory allergens.”

    We would like to thank the Reviewer for very useful comment.

    In our research Institute, we use animal strains depends on aim of study, nature of chemical and route of exposure. BALA/c mice are used for detection of immune responses to examine inhalation exposure to volatile organic compounds such as toluene, diesel exhaust particles and secondary organic aerosol. C3H mice are used for detection of respiratory responses in allergic animal models to examine chemical sensitivity reactions such as endocrine disruptors, phthalates, flame retardants and pesticides.

    Therefore, in the present study, as previously reported our published papers (Win Shwe et al., J Appl Toxicol. 2013, 33,1070-1078; Yanagisawa et al., J Appl Toxicol. 2016, 36, 1496-1504; Yanagisawa et al., 2018, J Immunotoxicol, 15, 31-40), we selected C3H mice to generate allergic asthmatic mouse models to detect the effects of BPA exposure.

On the other hand, C3H mice lack TLR4 and it has been reported that this TLR is important for memory. (2012) Evidence for a Developmental Role for TLR4 in Learning and Memory. PLoS ONE 7(10): e47522. This paper concludes: “Our findings suggest that TLR4 has a developmental role in shaping spatial and contextual learning and memory.”

  We would like to thank the Reviewer for invaluable comment.

    We agree that TLR4 is important for memory functions. Our research group have shown that involvement of TLR4 in pesticide diazinon-induced neurotoxicity using C3H/HeN (TLR4 intact) and C3H/HeJ (TLR4 deficient) adult male mice (Win Shwe et al., J UOEH 2012, 34; 1-13). We have also reported that role of TLR4 in olfactory-based spatial learning activity of neonatal mice after perinatal exposure to diesel exhaust origin secondary organic aerosol (Nway et al., Neurotoxicology, 2017, 63; 155-165).

   TLR4 knockout mice (TLR4-/-), TLR4 deficient mice (e.g., C3H/HeJ) and mice with inhibition of TLR4 by antagonists show different spatial learning and cognitive functions. It maybe due to involvement different brain regions, type of learning and memory function test (e.g. Morris water maze task, novel object recognition task) and different origin of animal strains.

How does the author show that the memory problems he observes are not due to the absence of TLR4?

   In our present study, we have compared the behavioral test with the control group (Vehicle-exposed group) with BPA-exposed groups with or without OVA immunization in C3H/HeJJcl mice. Therefore, our results are not due to the absence of TLR4.

The discussion is very long and does not really discuss the results.

    As commented by the Reviewer, we deleted unnecessary sentences in our revised manuscript.

Round 2

Reviewer 1 Report

The authors addressed all my concerns.

Minor comments:

Page 11 lines 39-40. I cannot understand the meaning of this sentence: “Our findings suggest that high dose of BPA alone COULD NOT INFLUENCE  memory function and COULD INFLUENCE in AA groups.” Please consider to rephrase this statement

Page 12 lines 37-38. Consider to rephrase as follows: “…sample size of ANIMALS AND lack of behavioral assessment…”…..”In our future study, we WILL increase the number of animals AND perform…”

Author Response

Responses to the Reviewer #1

We really appreciate the Reviewer for invaluable comments, suggestion and guidance for improvement of our manuscript.

Minor comments:

Page 11 lines 39-40. I cannot understand the meaning of this sentence: “Our findings suggest that high dose of BPA alone COULD NOT INFLUENCE  memory function and COULD INFLUENCE in AA groups.” Please consider to rephrase this statement

   Our findings suggest that high dose of BPA impaired memory function in allergic asthmatic individuals, however, it did not affect memory function in normal individuals.

Page 12 lines 37-38. Consider to rephrase as follows: “…sample size of ANIMALS AND lack of behavioral assessment…”…..”In our future study, we WILL increase the number of animals AND perform…”

   As suggested by the Reviewer, we corrected that sentence to “The limitations of our study are small sample size of animals and lack of behavioral assessment in BPA-L, BPA-M-exposed groups. In our future study, we will increase the number of animals and perform behavioral test in different doses of BPA-exposed mice.” in our revised manuscript.

Reviewer 3 Report

The author has answered my comments satisfactorily

Author Response

Responses to the Reviewer #3

The author has answered my comments satisfactorily.

   We really appreciate the Reviewer for invaluable comments, suggestion and guidance for improvement of our manuscript.

This manuscript is a resubmission of an earlier submission. The following is a list of the peer review reports and author responses from that submission.

Round 1

Reviewer 1 Report

In Figure 1 and Figure 2, the exploration time to familiar and novel objects in OVA and OVA+BPA-H groups are different, respectively. Which one is correct?

In Figure 1, the exploration time to the familial object in OVA+BPA-H group is higher than vehicle group. Conversely, the exploration time to the novel object in OVA+BPA-H group is lower than vehicle group. Please explain why there are such different phenomena and mechanism.

Please also describe and test the differences of exploration time and discrimination index of vehicle, OVA, BAP-L, OVA+BPA-L, BAP-M, OVA+BPA-M respectively. The authors can use a table instead of a graph to show the results.

The Kruskal-Wallis test is not a post-hoc analysis (Page 4, line 44-45). The Kruskal-Wallis test is a nonparametric test, and is used when the assumptions of one-way ANOVA are not met. If the Kruskal–Wallis test is significant, a post-hoc analysis can be performed to determine which levels of the independent variable differ from each other level. The post-hoc analysis is probably not used to confirm the significance in the related results.

Due to the small sample size, it is not correct to use the student’s t test to examine the differences of exploration time to novel object and familial object and mRNA expressions and body and brain weight between OVA () and OVA (+) groups, respectively (Page 4, line 45-47).

It is recommended that the authors should find a statistician to conduct data analysis. Please carefully check and modify the results of this article.

Author Response

Responses to the Reviewer #1

We would like to thank the Reviewer for invaluable suggestions and comments. According to the Reviewers’ comments, we have revised our manuscript thoroughly and highlighted by blue color. We have made Native English Check by International Medical Information Center, Tokyo (https://www.imic.or.jp/english/) and highlighted by underline.

In Figure 1 and Figure 2, the exploration time to familiar and novel objects in OVA and OVA+BPA-H groups are different, respectively. Which one is correct?

    The reason for different exploration times in the two figures for OVA-BPA-H are due to separate experiments. Please kindly see the explanation in next comment.

In Figure 1, the exploration time to the familial object in OVA+BPA-H group is higher than vehicle group. Conversely, the exploration time to the novel object in OVA+BPA-H group is lower than vehicle group. Please explain why there are such different phenomena and mechanism.

     The exploration time to the familial object in OVA+BPA-H group is higher than vehicle group, conversely, the exploration time to the novel object in OVA+BPA-H group is lower than vehicle group. It may be due to vehicle group expressed more interest in novel object than familial object and OVA+BPA-H group expressed little interest in novel object than familial object.

     We provided these issues in our Result section in our revised manuscript.

Please also describe and test the differences of exploration time and discrimination index of vehicle, OVA, BAP-L, OVA+BPA-L, BAP-M, OVA+BPA-M respectively. The authors can use a table instead of a graph to show the results.

 Firstly, we have performed the novel object recognition test in OVA-immunized allergic asthmatic mouse models. We used 5 groups including vehicle, OVA, OVA+BPA-L, OVA+BPA-M and OVA+BPA-H. As a result, OVA+BPA-H exposed male mice showed a significant decreased exploration time to novel object compared to familial object (P < 0.05, Figure 2A). Discrimination between familiar and novel objects was calculated using a DI. We found that the DI in BPA-H exposed mice was significantly reduced compared with that in the vehicle with OVA-exposed mice (P < 0.05, Figure 2B). These findings indicate that the mice exposed to OVA+BPA-H could not discriminate a novel object from a familiar object. These prompted us to investigate whether those effects are due to OVA or BPA. Thus, we conducted next experiment of novel object recognition using 4 groups including vehicle, BPA-H alone, OVA alone and OVA+BPA-H (Figure 3). Our result showed that co-exposure to BPA-H and OVA affects object recognition memory (P < 0.05, Figure 3B). Taken together, BPA-H alone does not have effects on novel object recognition ability and can affect in allergic asthmatic mice.

     We provided these issues in our Result section in our revised manuscript.

The Kruskal-Wallis test is not a post-hoc analysis (Page 4, line 44-45). The Kruskal-Wallis test is a nonparametric test, and is used when the assumptions of one-way ANOVA are not met. If the Kruskal–Wallis test is significant, a post-hoc analysis can be performed to determine which levels of the independent variable differ from each other level. The post-hoc analysis is probably not used to confirm the significance in the related results.

Due to the small sample size, it is not correct to use the student’s t test to examine the differences of exploration time to novel object and familial object and mRNA expressions and body and brain weight between OVA (‑) and OVA (+) groups, respectively (Page 4, line 45-47).

It is recommended that the authors should find a statistician to conduct data analysis. Please carefully check and modify the results of this article.

  As suggested by the Reviewer, we have discussed with Statistician in our Research Institute and we revised our statistical analyses as follows;

All the data were expressed as the mean ± standard error (S.E.). The statistical analysis was performed using the StatMate II statistical analysis system for Microsoft Excel, Version 5.0 (Nankodo Inc., Tokyo, Japan). The data were analyzed using a one-way analysis of variance with a post-hoc analysis using the Bonferroni/Dunn method. Differences were considered significant at P < 0.05.

Reviewer 2 Report

I would like to thank the editor for giving me the opportunity to review this paper again. In this interesting manuscript, the authors aimed to evaluate potential changes induced by high exposure BPA in allergic asthmatic mice on both memory function and neuroimmune biomarkers. The manuscript was considerably improved and enriched with immunohistochemical analysis, new figures and a table. Introduction contains sufficient elements that characterize background and properly identify purposes of the study, whereas discussion was significantly ameliorated. This study provides new insights in  health effects associated to BPA exposure, exploring a field still partially unknwn, and opening up new scenarios for further investigations in animals and humans. Nonetheless, before to be considered for publication, I believe that the manuscript needs some minor revisions.

General comments.

I recommend the authors a revision of English, in addition to a check of the correct spacing throughout the text.

A list of abbreviations is also suggested.

Specific comments

Abstract:

Line 23: “were” and not “was”

Line 28: please delete comma after “and”

Line 31: consider to delete “which”

Introduction:

Page 1, line 40: “routes” and not “route”.

Page 1, line 43 – page 2, lines 1-3: consider to join the two sentences.

Page 2, lines 20, 29, 42: consider to write “A recent study….”

The full name of TNF should be written at line 34 and not at line 51 of page 2.

Page 2, lines 34, 35, consider to amend as follows: “AN animal study”, “A human study”

Page 2, line 6: “..plasma IL-6 LEVELS were found…”

Page 2, line 46: please delete comma after “and”.

Page 2, lines 47-48: consider to rephrase the sentence ad follows: “However, the effects of chemical BPA exposure on COGNITIVE FUNCTION AND NEUROINFLAMMATOY RESPONSES OF ASTHMATIC SUBJECTS…”

Methods:

Page 4, lines 3-5: consider to rephrase this statement since it is rather unclear.

Page 4, line 7: the authors wrote that the animals were sacrificed before the last OVA administration, however above (page 3, lines 26-27) they stated “….recognition test was conducted before the final administration of OVA, hypothalamus was collected later…”. Could they clarify this specific point?

Page 4, line 8: consider to amend as follows: “…the hypothalamus AND HYPPOCAMPUS WERE collected…”

Results:

In the title “novel object recognition test” was repeated twice.

Figure 3: I suggest the authors to use the same colors used in figure 2, namely red for the familial object and blue for the novel object. In addition, why are the exploration time are so different in the two figures for OVA-BPA-H?

Page 8, line 12: based on Figure 4c, the levels of expression receptor subunits NR2B were reduced, and not increased, in OVA+BPA‑H mice.

Discussion:

In this section, there is an excessive use of the phrases “the present study” and “accompanied with”..

Page 10, lines 19: please correct as “A common route….”

Page 10, line 27: “routes” and not “route”. Please note that the paragraph from line 19 of page 10 to line 2 of page 11 contains repetitions about the existence of different pathways of exposure to BPA, thus consider to join them in a single sentence.

Page 12, line 4: “is” and not “are”

Author Response

Responses to the Reviewer #2

We would like to thank the Reviewer for invaluable suggestions and comments. According to the Reviewers’ comments, we have revised our manuscript thoroughly and highlighted by blue color. We have made Native English Check by International Medical Information Center, Tokyo (https://www.imic.or.jp/english/) and highlighted by underline.

General comments

I recommend the authors a revision of English, in addition to a check of the correct spacing throughout the text.

  As commented by the Reviewer, we have made Native English Check by International Medical Information Center, Tokyo (https://www.imic.or.jp/english/).

A list of abbreviations is also suggested.

  As suggested by the Reviewer, we added “List of abbreviations” in our revised manuscript as follows;

List of abbreviations

ABI       : Applied Biosystems

BPA       : Bisphenol A

CNS      : central nervous system

COX2   : cyclooxygenase-2

DEHP   : di-(2-ethylhexyl) phthalate

DI          : discrimination index

ER         : estrogen receptor

HPA      : hypothalamic-pituitary-adrenal

Iba-1      : ionized calcium binding adapter molecule 1

IL          : interleukin

NGF      : nerve growth factor

NMDA  : N-methyl-D-aspartate

OVA      : ovalbumin

oxtr        : oxytocin receptor

PBS       : phosphate buffered saline

RT‑PCR             : reverse transcription polymerase chain reaction

S.E.       : standard error

TNF      : tumor necrosis factor

TrkA      : tropomyosin receptor kinase A

Specific comments

Abstract:

Line 23: “were” and not “was”

   We corrected to “were” in our revised manuscript.

Line 28: please delete comma after “and”

   We deleted “comma”.

Line 31: consider to delete “which”

   We deleted “which”.

Introduction:

Page 1, line 40: “routes” and not “route”.

  We corrected to “routes”.

Page 1, line 43 – page 2, lines 1-3: consider to join the two sentences.

   As suggested by the Reviewer, we combined two sentences like “Perinatal exposure to BPA not only enhanced lung inflammation in the presence of allergen [4] but also induced aggressive and anxiety behaviors in adult rats [5] and altered brain sexual differentiation of various brain regions in rodents [6, 7].” in our revised manuscript.

Page 2, lines 20, 29, 42: consider to write “A recent study….”

    We corrected to “A recent study…”

The full name of TNF should be written at line 34 and not at line 51 of page 2.

    We described full name of TNF.

Page 2, lines 34, 35, consider to amend as follows: “AN animal study”, “A human study”

    We corrected to “An animal study” and “A human study”.

Page 2, line 6: “ plasma IL-6 LEVELS were found…”

    We corrected to “plasma IL-6 levels were found….”

Page 2, line 46: please delete comma after “and”.

    We deleted comma.

Page 2, lines 47-48: consider to rephrase the sentence ad follows: “However, the effects of chemical BPA exposure on COGNITIVE FUNCTION AND NEUROINFLAMMATOY RESPONSES OF ASTHMATIC SUBJECTS…”

    As suggested by the Reviewer, we re-write that sentence to “However, the effects of chemical BPA exposure on cognitive function and neuroinflammatoy responses of asthmatic subjects are largely unknown (Figure 1).

Methods:

Page 4, lines 3-5: consider to rephrase this statement since it is rather unclear.

     As commented by the Reviewer, we corrected that sentence to “At 11 weeks of age, the mice (n = 5~6 from each group) were sacrificed under deep pentobarbital anesthesia and the hippocampus and hypothalamus were collected……”.

Page 4, line 7: the authors wrote that the animals were sacrificed before the last OVA administration, however above (page 3, lines 26-27) they stated “….recognition test was conducted before the final administration of OVA, hypothalamus was collected later…”. Could they clarify this specific point?

Page 4, line 8: consider to amend as follows: “…the hypothalamus AND HYPPOCAMPUS WERE collected…”

   We corrected to “…..the hippocampus and hypothalamus were collected….”

Results:

In the title “novel object recognition test” was repeated twice.

    We deleted unnecessary one in Figure 2 legend in our revised manuscript.

Figure 3: I suggest the authors to use the same colors used in figure 2, namely red for the familial object and blue for the novel object. In addition, why are the exploration time are so different in the two figures for OVA-BPA-H?

    As suggested by the Reviewer, we used same colors in Figure 2 and 3, namely red for the familial object and blue for the novel object in our revised manuscript.

The reason for different exploration times in the two figures for OVA-BPA-H are due to separate experiments. Firstly, we have performed the novel object recognition test in OVA-immunized allergic asthmatic mouse models. We used 5 groups including vehicle, OVA, OVA+BPA-L, OVA+BPA-M and OVA+BPA-H. As a result, OVA+BPA-H exposed male mice showed a significant decreased exploration time to novel object compared to familial object (P < 0.05, Figure 2A). Discrimination between familiar and novel objects was calculated using a DI. We found that the DI in BPA-H exposed mice was significantly reduced compared with that in the vehicle with OVA-exposed mice (P < 0.05, Figure 2B). These findings indicate that the mice exposed to OVA+BPA-H could not discriminate a novel object from a familiar object. These prompted us to investigate whether those effects are due to OVA or BPA. Thus, we conducted next experiment of novel object recognition using 4 groups including vehicle, BPA-H alone, OVA alone and OVA+BPA-H (Figure 3). Our result showed that co-exposure to BPA-H and OVA affects object recognition memory (P < 0.05, Figure 3B). Taken together, BPA-H alone does not have effects on novel object recognition ability and can affect in allergic asthmatic mice. We described this issue in Result section under 3.2. Effect of BPA Exposure on Novel Object Recognition Test in our revised manuscript.

Page 8, line 12: based on Figure 4c, the levels of expression receptor subunits NR2B were reduced, and not increased, in OVA+BPA‑H mice.

  We corrected “Significantly reduced mRNA expression level of the NMDA receptor subunits NR2B was found in the hippocampi of mice exposed to OVA+BPA-H compared to the control (P < 0.05, Figure 4C).”

Discussion:

In this section, there is an excessive use of the phrases “the present study” and “accompanied with”.

    We edited this issue in our revised manuscript.

Page 10, lines 19: please correct as “A common route….”

   We corrected to “A common route…”.

Page 10, line 27: “routes” and not “route”. Please note that the paragraph from line 19 of page 10 to line 2 of page 11 contains repetitions about the existence of different pathways of exposure to BPA, thus consider to join them in a single sentence.

    We corrected “route” to “routes”.

    We have revised repetitions in our revised manuscript.

Page 12, line 4: “is” and not “are”

   We have revised that sentence.

Round 2

Reviewer 1 Report

The statistical methods used in this article are still unclear and incomplete. Inappropriate statistical methods can affect the significance of the results. It is necessary to seek a professional statistician to conduct data analysis and revise the results.

1. In Figure 2 and Figure 3, the exploration time to familiar and novel objects in OVA and OVA+BPA-H groups are different, respectively. Which one is correct?

The authors response that “The reason for different exploration times in the two figures for OVA-BPA-H are due to separate experiments. Please kindly see the explanation in next comment.”

(1) In the section of 2.2 experimental design, readers can’t know they are separate experiments.

(2) The experimental results of the exploration time to familiar and novel objects in BPA-L and BPA-M groups are not shown in the article.

(3) In figure 1, the exploration time to familiar and novel objects in the OVA+BPA-H group are significantly different. However, in figure 2, there are no statistical different between exploration time to familiar and novel objects in the OVA+BPA-H group. Thus, there are no enough evidence to show “co-exposure to BPA‑H and OVA affects object recognition memory”. Obviously, there are other variables that affect the novel object recognition test.

2.The authors still do not perform the post-hoc analysis. The Kruskal-Wallis test is a nonparametric test, and is used when the assumptions of one-way ANOVA are not met. If the Kruskal–Wallis test is significant, a post-hoc analysis can be performed to determine which levels of the independent variable differ from each other level. In the present article, the post-hoc analysis is not used to confirm the significance in the related results (fig. 2B, fig. 2C, fig 4C, fig 5). For example (figure 2B), if the Kruskal–Wallis test is significant, a post-hoc analysis is used to further compare the significant differences of DI between vehicle, OVA, OVA+BPA-L OVA+BPA-M, and OVA+BPA-H groups, respectively. There are no p values from the post-hoc test that shown in the results.

3.The authors didn’t response the last reviewed question.  

Due to the small sample size, it is not appropriate to use the student’s t test to examine the differences of exploration time to novel object and familial object and mRNA expressions and body and brain weight between OVA (‑) and OVA (+) groups, respectively.

4. Please perform and describe the correct statistical methods in the present article.